# Sustainable Local Exploitation and Innovation on Meat Products Based on the Autochthonous Bovine Breed *Jarmelista*

Paula Coutinho [1], Manuel Simões [2], Carlos Pereira [3,4] and Teresa Paiva [1,5,6,*]

1   Center of Potential and Innovation of Natural Resources, Polytechnic Institute of Guarda,
    6300-559 Guarda, Portugal; coutinho@ipg.pt
2   InnovPlantProtect Collaborative Laboratory, Estrada de Gil Vaz, Apartado 72, 7350-999 Elvas, Portugal;
    manuel.simoes@iplantprotect.pt
3   Polytechnic of Coimbra-School of Agriculture, 3045-601 Coimbra, Portugal; cpereira@esac.pt
4   CERNAS—Research Centre for Natural Resources, Environment and Society, 3045-601 Coimbra, Portugal
5   Technological and Management School, CI&DEI, Guarda Polytechnic Institute, 6300-559 Guarda, Portugal
6   Research Center in Business Sciences, University of Beira Interior, 6200-609 Covilhã, Portugal
*   Correspondence: tpaiva@ipg.pt

**Abstract:** The Jarmelista autochthonous bovine breed has a sustainable production and is part of the culture of the Portuguese territory, representing a touristic attraction and originating a differentiated beef product that can only be found in a particular region of the country. However rural and livestock population evolution in Portugal's inland has demonstrated a great regression with consequences for environment and nature conservation. In this context and considering that silvopastoral activity has shaped the natural areas of mountain territories since its beginning, rethinking the importance of such activity has become vital for the territory sustainability. In this perspective, this work presents an analysis of the adaptation and evolution of Jarmelista bovine breed production to current times, perceiving its limitations, challenges, and success potential, supported by a data collection of secondary and primary sources. Despite the natural, healthy, and sustainable value of this particular bovine meat, we observed that is still not recognised by the market or even by the producers. The inability of proving the Jarmelista beef added value within the value chain is the main cause of businesses and consumers sceptic and disbelief in the potential of its economic and tourism contribution. Several possibilities and actions were identified to contradict this path.

**Keywords:** autochthonous Jarmelista bovine breed; sustainable production; beef added value; territorial valorisation

## 1. Introduction

Euro-Mediterranean mountains have a long history of co-evolution with human activities and can be considered agroecosystems (mostly as grazing livestock systems). However, there are evidences of local breeds suffering intensely with the abandonment and intensification of agriculture, and the situation is especially critical in Europe. In many cases, the continuation of traditional farming practices is determinant for the maintenance of the biodiversity value [1], the ecosystem services [2], or the protection against natural hazards [3,4]. Nowadays, arguments for conserving low-yielding local breeds are widely recognised by the scientific community, public administration, farmers and public in general, once is highlighted that autochthonous breeds produce locally ecosystem services, provisioning food, but also cultural services, such as maintenance of cultural heritage or identity and regulating services such as landscape and biodiversity management [5].

Agriculture as a provider of food, fibre and shelter to the human population is a sector that plays a determinant role in moving towards sustainable development [6]. The majority of authors and international organisations agree that food sufficiency, environment preservation, socio-economic viability and equity are important sustainability components (e.g., [7–10]). Nevertheless, one of the biggest challenges for sustainable development

is the definition of operating models that can explicitly demonstrate the environmental, economic, and social advantages and disadvantages of the different production systems and strategies as part of a common analysis framework [9].

According to De Barcellos, et al. [11], European consumers have developed a preference for biological practices and organic production methods, while disapproving genetic modification and excessive food processing [12]. The growing trend in biological food consumption continued in 2011 despite the economic crisis [13]. The European biological food market is emerging from its pioneering phase and, in some countries, it is growing at "maturity". Organic retail sales in Europe were valued at 40.1 billion euros with a recorded growth trend of 10% per year. Between 2009–2018, the biological market in Europe more than doubled [14]. According to Willer, et al. [15], the demand for biological food in Europe has almost doubled in the first decade of this century. Meat is ranked in third place in the categories of biological products that are best sellers, representing 10% of sales in EU markets [16]. In Portugal, biological agriculture has developed considerably since its inception in 1985. In 2018, there were 2820 biological farms cultivating 239,860 hectares (ha), which correspond to 6% of total agricultural land. Pastures occupy almost 70% of this area, and almost 63% of Alentejo district area is dedicated to organic farming. In the country, almost 100,000 bovines are reared under this regime [17]. However, animal breeding in biological production is not yet well developed, and the production volume is still meagre. Portugal is not self-sufficient in beef. However, it is possible to appreciate a market for beef produced by autochthonous breeds under extensive and organic production systems. Beef from autochthonous breeds is considered a high-quality product, mainly because of the unique taste and texture, resulting from the production system involving slow growth rate and feed grazing. This product differentiation has allowed the enlargement of small niche markets and has led to an increase in beef value with a Protected Designation of Origin (PDO) or sold as organic beef [18]. Since Portuguese agriculture cannot compete on quantity or production cost with other competitors, differentiation and quality seem to be the alternatives that may stimulate rural activities in Less Favoured Areas (LFA) and create a regional benefit able to contribute to sustainable development. Extensive animal production systems can be an essential component of environmental and landscape protection and contribute to invert the decrease of the human and the physical desertification of our rural areas and animal genetic resources [19].

Local breeds are envisaged as an essential alternative to farmers from the LFA, contributing to farmers' income, rural development, and landscapes conservation. However, in some cases, the lack of notability of the meat produced in a sustainable regimen and organic production, as well as its importance to biodiversity preservation, do not allow the recognition of its potential. This fact is particularly noticed in autochthonous breeds with small flocks. In these cases, producers lack the dimension to organise meat distribution and rely solely on local markets. Therefore, the development of differentiated and high-quality products with high added value seems to be the only alternative to stimulate these regions' rural activity. Furthermore, consumers demand for quality meat is increasing, and, in Portugal, beef production from local breeds can carry out these properties [20]. It is important to stress that socio-economic driven land-use changes, namely regarding the effects of abandonment, will have a more significant influence than climate change on ecosystem services until the middle of the century, and trajectories must be well-defined, especially in mountain regions [21].

From the exposed, it is pertinent to study the rearing system of the autochthonous Jarmelista bovine breed and its relevance to the territorial economy and sustainability. Among the set of economic, social, and environmental parameters, a critical one is the contribution of these systems to the fight against mountain areas' human abandonment by providing added value in economic and socio-environmental terms. These systems need revitalisation, by improving their profitability and promoting the rejuvenation of the farming population and contributing to biodiversity preservation, by dealing with breeds

of high rusticity, which can be envisaged as natural transformers of intrinsic resources of the mountain areas.

## 2. Materials and Methods

In Portugal, there is a population of 20 farmers rearing cattle of the Jarmelista breed (Census 2019), of which some, additionally, also devote themselves to the production of beef that does not belong to this autochthonous breed. In order to characterise the livestock production system (pasture system, organisation, and productivity) and how it is related to the sustainability of the territory, two information sets were needed. First, an information collection on the Jarmelista breed system's biodiversity connection and the territory characteristics was carried out based on previous studies and literature review. Data regarding the Jarmelista carcass characteristics were also evaluated based on registers referring to the years 2001–2018. In this way, it was possible to better understand and discuss the results of the second type of data collection of the 20 farms responsible for the production of Jarmelista breed, through a direct and structured survey conducted by Acriguarda (Meat Farmers Regional Association), concerning the characterisation and reproductive performance of the Jarmelista breed (see Appendix A). Although the sample size seems small in absolute terms, it was highly representative of the target population, once represents the totality of farmers producing Jarmelista in the Guarda county. The data were coded, entered a database, and reviewed by the research group, before being submitted to statistical analysis through Stat Soft Statistica™ 8.0 (Statsoft Inc., Tulsa, OK, USA). A descriptive analysis of the data collected was performed and summarised in tables and graphs.

## 3. Results

### 3.1. Portuguese Agro Biodiversity

Despite its small dimensions, Portugal has a huge environmental variability of orography conditions, soils, climate, land structure, social and cultural traditions. Furthermore, Portugal has great biodiversity, particularly regarding officially recognised as autochthonous breeds (15 bovine, 16 ovine, six caprine, three swine, six equine and four chickens' breeds). These breeds and their exploitation represent an important part of the country's historical and cultural heritage, contributing to the settlement of population in rural areas and resulting in different manifestations of gastronomy, social and cultural traditions [22]. Consequently, agriculture and forestry are crucial for sustainable development in the Guarda region in its economic, social, and environmental aspects. According to INE (2009) [23], the Utilised Agricultural Area (UAA), represents 29.15% of Guarda district territory which corresponds to 161,405 ha, of which, permanent pastures occupy almost 50%. In Guarda district, 65.45% of the UAA (about 2/3 of total) is used for livestock production, highlighting the great importance of this sector and its impact in the economy and the region's society. The same source also indicates that existed 30,375 bovines in the region, 153,348 sheep, 20,403 goats, 13,759 pigs, 4585 equines, among other farm animals (INE 2009). Several autochthonous breeds originated in this region, namely: the goat's breed "Cabra Serrana" ("Cabra Jarmelista", "Cabrito da Beira" (Geographical Protected Indication—GPI)), the sheep breeds "Churra Mondegueira" and "Bordaleira Serra da Estrela" (producers of milk used for one of the most important PDO cheese of the country, "Queijo Serra da Estrela"), "Borrego Serra da Estrela" (PDO), the Serra da Estrela lamb meat, and the bovine breed "Jarmelista". In the last years, a significant decrease in the number of animals from autochthonous breeds in the livestock production was registered in the region and as stated by other authors, reducing the number of effective from autochthonous breeds would necessarily result in the desertification, with an emergency of actions in order to add value to this sector and products [22].

Globally, in animal production, meat still occupies a place of crucial importance, and traditionally, beef is the most relevant in the market. In 2013, Portugal had the 14th largest

cattle herd in the EU, with approximately 1,471,000 heads which remained constant over the years, contrary to other kinds of livestock for which the trend was negative [10,24,25]

The bovine production in Portugal has traditionally been carried out on farms specialised in the farming or rearing and fattening [26]. In 2018, beef production represented about 20% of the total volume of meat produced in Portugal, and the certified meat (PDO and Protected Geographical Indication-PGI) represented only a 2.6% of the total beef production [27]. With the bovine spongiform encephalopathy (BSE) crisis in the late 1990s and early 2000s, beef consumption declined. However, according to Ralo [28], the BSE crisis brought autochthonous breeds back into line due to their resilience to disease and consumers' confidence. Climate aggression gives these breeds so-called rusticity and significant competition power [29]. According to GPP (2017) [30], beef consumption in Portugal has been decreasing since 2008. In that same year, beef consumption was 19.6 kg/hab.; in 2013, consumption decreased to 16.9 kg/hab. INE [23].

In terms of value, domestic consumption of beef in Portugal remained stable in 2008 and 2009 representing 207 thousand tons, decreasing in the following years by approximately 15% between 2009 and 2013 [23]. The production of beef in Portugal is not enough for domestic demand, and this situation has been worsening in recent years, according to [27]. From 2009 until 2012, the degree of self-supply only covered 52% of consumption needs. This situation leads the country to a noticeable dependence on this product, placing Portugal in a very vulnerable situation to price fluctuations in the international market.

Santos [31] concluded that meats are substitute goods for each other, which means that when the price of one meat increases, others' demand also increases. Therefore, the consumption of meat in the market at the best price increases, regardless of being a national or imported product. Considering the national meat market price, Portugal may find it challenging to compete in an undifferentiated market, where low price strategies predominate. In a context of increasing trade globalisation, where Portugal competes in the national market with European Union partners and, with large world exporters, there are several difficulties in lowering prices due to the reduced margin concerning production costs. This may have adverse consequences for Portuguese animal production.

In 2006 a study was developed to characterise morphologically Jarmelista breed. 185 morphological characters were used for females and 170 characters for males using data analysis/numerical taxonomy methods. Simultaneously, with the collection and recording of morphological data, blood, hair, and semen for DNA analysis, as well as the images of all animals identified as Jarmelista, this breed was compared with other autochthonous bovine breeds of Portugal. Moreover, the study revealed that Jarmelista constitutes a distinct group and independent of all the already recognised autochthonous breeds [32]. Hence, this breed was recognised as a native breed since 27 of October of 2007, with an initial number of 34 registered animals. Its genealogical book has been taken over by Acriguarda, the breeders association.

As previously mentioned, Jarmelista farms are found in a characteristic mountain region with climatic conditions registering high thermal amplitudes, which have strong implications on pasture development and animal's growth. The low number of animals per holding characterises the breeding regimen of these animals. Jarmelista breed is characterised by the animals' rusticity, perfectly adapted to the severe climatic conditions of the region; compared to other animals raised in the region this breed demonstrates greater strength and robustness [22]. Furthermore, animals are fed with natural pastures, oats, rye (straw and grain), herb fodder and hay, being consequently produced in a sustainable and organic regimen.

### 3.2. Characterisation of the Territory

Guarda district is crossed by a vast mountain range called Central System that incorporates about 85% of Serra da Estrela's total surface in Portugal. This region presents a sharp relief, from 84 m in Vila Nova de Foz Côa to 1993 m in Serra da Estrela [33]. Guarda county is the coldest region, and the low temperatures registered on winter affect pastures

development dramatically [22]. On average, temperatures from December to February in Guarda are below 5 °C, which hinder vegetation growth. Furthermore, below 10 °C grass growth is low, and only after April the temperatures rise above 10 °C. Legumes need higher temperatures than grasses to grow because higher temperatures lead to increased photosynthesis. The ideal temperatures for pastures based on grasses to grow lie between 20 and 25 °C [34].

Besides, the rainfall in Guarda is irregular, with an annual average of 914 mm [35] and follows the Mediterranean climate's characteristic distribution, with a dry summer and a wet winter [34]. The lack of water is another main restriction on pasture production, leading to a lack of production from June to September [36] and the hydric deficit also affects the nutritional value of vegetation [37]. The low thickness of the soil results in low water retention capacity, worsening the effect of rainfall seasonality. On the other hand, soil's soaking tendency compromises the development of some species and the management of bovines. The soils of Guarda region generally have a granitic origin, coarse texture, and acid pH, which constitutes one of the main limiting factors in the development of altitude pastures, high potassium levels and low to medium levels phosphorus and organic matter. Except for sludge, the soil's thickness is significantly reduced, which combined with a low herbaceous cover, coarse texture, rugged relief, and nature of rocky materials, lead to drainage problems in some cases and high susceptibility to erosion in others [35].

The soil and climatic characteristics of a mountainous region with skeletal soils submitted to erosion, and labour shortage, led to a larger and predominant number of extensive livestock farms with a low number of animals. Highland pastures, which are fundamental for this region's economic sustainability and for these animals, consist essentially of the traditional "lameiros". Lameiros are natural pastures found in altitude lands that grow without the interference of man. They are preferably located near waterways or naturally moist areas, benefiting from irrigation in whole or in part, thus occupying the best soils [34]. Some pastures are sown and improved with herbaceous species adapted to the region's edaphic conditions, rye, and oat cereal pastures. Lameiros are permanent natural pastures dominated by nearly perennial grassy natural plants, some of which are very fibrous. Improvement of mountain pastures in quantity and quality involves sowing improved pulses and grass species and fertilising them. Increasing the productivity of these ecosystems, which are fundamental for the protection of soil and water, will increase the income level of the populations, an essential condition for their settlement in the region combating human desertification [35].

### 3.3. Characterisation of the Production System

The set of data obtained through the questionnaire gathered information from 20 producers of the Jarmelista breed in which 14 breeders were from Guarda Municipality (from the parishes of Guarda: 10%; S. Pedro do Jarmelo 20%; and Miguel do Jarmelo: 10%), three of the Municipality of Pinhel, one of the Municipality of Belmonte, one of the Municipality of Seia and 1 of the Municipality of Almeida (Figure 1).

The majority of the farms are from small dimension. The farms of the 20 producers could be divided in three classes. Four farms have more than 60 ha of total area, and four others have areas between 30 and 60 ha, while the remaining 12 have areas of less than 30 ha. Figure 2 presents the three classes of farms according to their total areas, utilised agricultural area (UAAs), pastures area and area of crops dedicated to cultures used for animal feed. As can be seen, pastures represent more than 50% of the farm areas in all cases. Furthermore, 95% of the breeders have less than 10 hectares of annual crops for livestock feed.

Regarding the feed system (Table 1), most of the farmers raised their animals without commercial feeds or other types of processed foods. However, it can be pointed out that two producers only use commercial feed. These two producers had 10 and 11 Jarmelista cows, respectively, and seven and eight Jarmelista calves with ages ranging from 6 to 24 months.

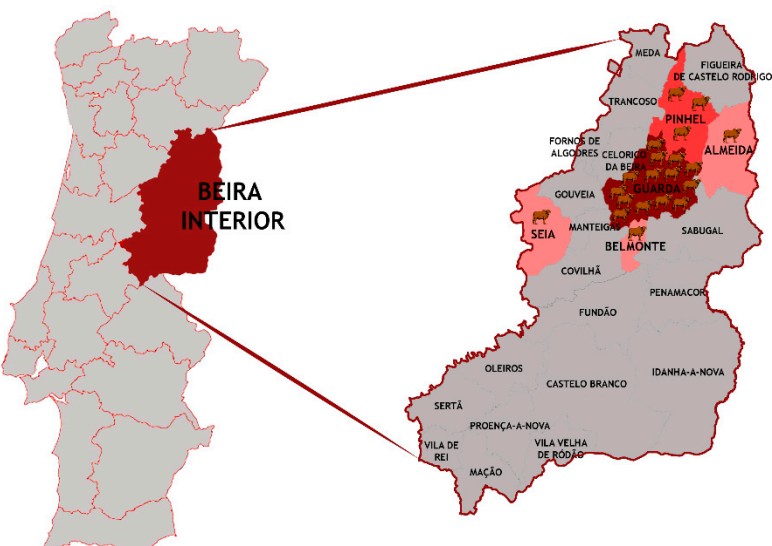

**Figure 1.** Map of the Region and Jarmelista productions. Source: Author's calculation from research dataset of structured survey.

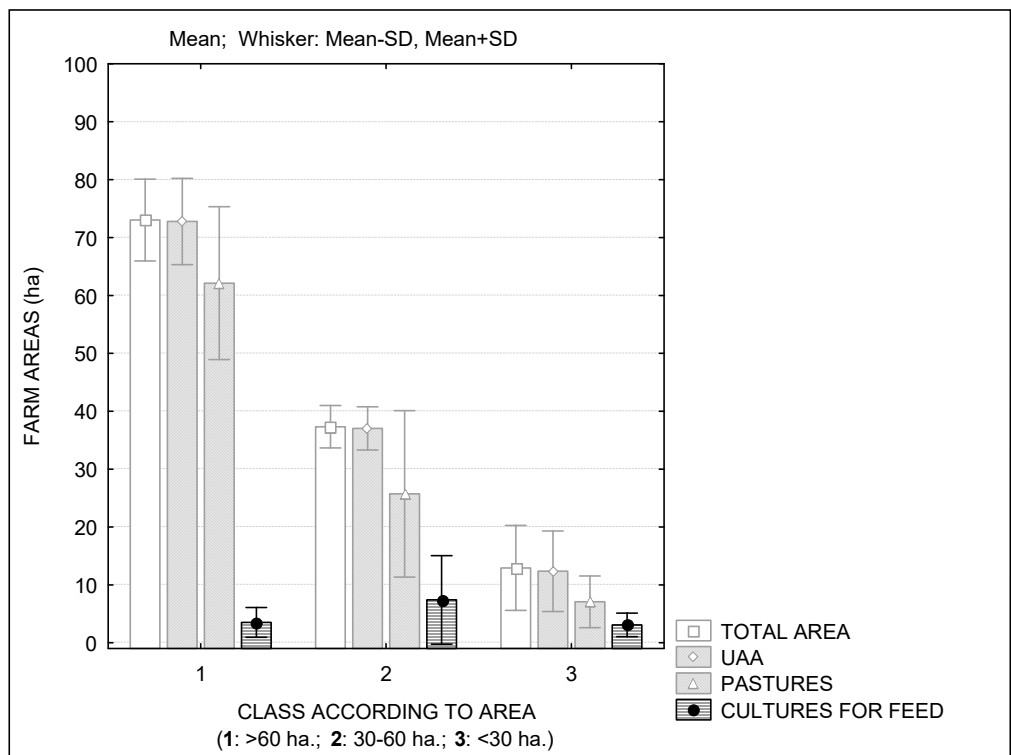

**Figure 2.** Characterisation of Jarmelista producer's farms. Source: authors' calculations from the structured survey research dataset.

By crossing the information regarding the age of the animals and their type of feed, it was found that, except for those two producers, the introduction of commercial feed, when it occurs, is only after the weaning or in adulthood. After weaning and until slaughter, 45% of the farmers inquired opted for a feeding regime composed of grazing, hay, and straw. It was also noticed that, of the different feeding options available (grazing, hay, straw, feed, oat fodder, rye fodder and herb fodder), 85% use grazing; 75% use hay; 70% use straw; 40% use feed; 10% use oats and rye forage; and 5% herb fodder, as noted in Table 1.

**Table 1.** Matrix of feeds used by the different producers of Jarmelista breed.

| Producer | Feeds Used | | | | | |
|---|---|---|---|---|---|---|
| | Grazing | Hay | Straw | Commercial Feed | Oat/Rye Fodder | Herb Fodder |
| 1 | X | X | | X | | |
| 2 | X | | X | X | | |
| 3 | X | | X | X | | |
| 4 | X | | | | X | X |
| 5 | | X | X | X | | |
| 6 | X | X | X | | | |
| 7 | X | X | | | X | |
| 8 | X | X | | | | |
| 9 | X | X | X | | | |
| 10 | X | X | X | | | |
| 11 | X | X | X | | | |
| 12 | X | X | X | X | | |
| 13 | X | X | X | | | |
| 14 | X | X | X | X | | |
| 15 | X | X | X | | | |
| 16 | X | X | X | | | |
| 17 | X | X | X | | | |
| 18 | X | X | X | | | |
| 19 | | | | X | | |
| 20 | | | | X | | |
| % Usage | 85 | 75 | 70 | 40 | 10 | 5 |

Source: authors' calculations from the structured survey research dataset.

As previously pointed, altitude pastures are subjected to specific limitations, namely their climate and topography. The availability and quality of pasture are not constant throughout the year, so it is not always sufficient to cover animals' needs throughout their growth cycle. Low temperatures and the lack of water, are the main limitations for pasture development. During this period, supplements such as hay and straw are provided. A correct and balanced diet is one of the most critical factors in cattle production and, therefore, pasture management, by allowing control over the quantity and quality of available grass, becomes crucial for the nutritive value of the grass, which in turn determines the animal diet.

The maintenance of this autochthonous breed in the region can have a significant contribution to sustainability either directly or indirectly. The maintenance of altitude pastures and associated agricultural practices contributes to avoid the abandonment of these areas and prevents the excessive accumulation of biomass, reducing therefore the potential impacts of forest fires. Besides, the maintenance of these pastures avoids the expansion of land areas dedicated to the production of *Eucaliptus globulus* which has been a matter of continuous controversy in Portugal due to its negative impact over native species. Therefore, extensive cattle grazing in the region has a positive impact on the environment and, simultaneously, can contribute to reduce the country's dependence on external meat markets. Exotic breeds are not well adapted to the harsh weather conditions of the region and require improved quality feed, namely the use of commercial feeds.

With the surveys, we were also able to evaluate some zootechnic parameters which are presented in Table 2. Farmers scarcely fatten the cattle, for more than 18 months, not only to avoid the competition for food but also due to the need for extra housing for the

younger animals, which can be a problem for some producers due to the small areas of the farm.

Regarding the reproductive management of the animals, it was possible to observe that, on average, the animals had the 1st delivery at the age of ca. 38 months. The youngest animal to have at the first delivery had 25 months and the oldest around 70 months.

**Table 2.** Reproductive management parameters evaluated from enquires.

| | Age at 1st Delivery (Months) | Interval between Deliveries (Days) | Fertility Rate | Age of Registered Females (Years) | Age of Registered Males (Years) |
|---|---|---|---|---|---|
| **Average ± SD** | 38.32 ± 10.36 | 440.28 ± 57.43 | 84.22 ± 10.42 | 6.6 ± 1.77 | 2.6 ± 2.18 |

Source: Author's calculation from research dataset of structured survey.

In Table 2 we can, also, see that the mean interval between deliveries is 440 days. The average fertility rate registered was 84%. Cows have an average age between 6 and 7 years.

Overall, Jarmelista animals represent ca. 27% of the flock owned by these farmers. Considering just the Jarmelista flock, cows represent 54.2%, while bulls represent 3.4%. Male Jarmelista calves from 6 to 24 months represent 20.2%, while female calves within the same age group represent 18.5%. The remaining animals correspond to female calves within the age group 6–12 months (2.4%) and male calves of the same age group (1.4%).

Concerning the total flock owed by the 20 farmers, 161 Jarmelista cows correspond to one-third of the cows in production. Jarmelista bulls correspond to 35.7% of the existing bulls. Jarmelista female calves with 6–24 months correspond to 42.6% of the female calves with the same age, while Jarmelista male calves correspond to 46.5% of total calves of the same age class. Jarmelista female and male calves with 6–12 months correspond to 58.3 and 57.2% of calves' total population with 6–12 months.

From the total number of animals sold by the farmers in 2016 (246), ca. 37% corresponded to the Jarmelista breed (91). Of the 157 live animals sold to other producers in 2016, 25.5% were Jarmelista, corresponding to a sales volume of around €24,000. Besides, it was possible to observe that the sale price of Jarmelista animals was identical to the other breeds. Of the 89 animals sold for slaughter, 57.3% were Jarmelista corresponding to a sales volume of € 30,600, being the price identical to the other animals sold for slaughter. Thus, Jarmelista animals either sold to other producers or for slaughter, were valued equally and at the same price of other breeds.

In terms of this breed's economic profitability, it is observed that the production of Jarmelo cattle is not significant in the overall income of farmers due to the lack of valorisation of its contribution to biological and sustainable farming. Furthermore, this lack of valorisation also results from the fact that this breed is not recognised as differentiated in the market. For most farms, Jarmelista breed represents just a complement to the exploitation mainly because of cultural reasons, history, and animals' beauty. Due to the slower development of this breed, exotic breeds or crusades are preferred.

In Figure 3, we can observe the weight of Jarmelista cattle's sale in farm income (sales of Jarmelista animals/total sales of animals), noting that there are farms (6 breeders, i.e., 30% of respondents) to whom this breed corresponds to the entire business. In comparison, others have not made any sale of animals (three) or have not sold Jarmelista animals (two) during the inquiry year. For seven farmers, the proportion of Jarmelista cattle sales in the total farm income represented from 5 to 30%, while for two producers it represented ca. 75%.

In what concerns the meat production performance of the Jarmelista breed, available data related to the carcass conformation of animals slaughtered between 2001 and 2018 is presented in Table 3. The data corresponds to a total of 338 Jarmelista carcasses of different age classes. Conformation (the shape and development of the carcass) is denoted by E, U, R, O, P, with E being the best and P the poorest. The majority of the carcasses (295) were graded as R or O. Letter R corresponds to a well-developed round and shoulder with thick

back. Letter O corresponds to average round, slightly lacking thickness on a marginally flat back.

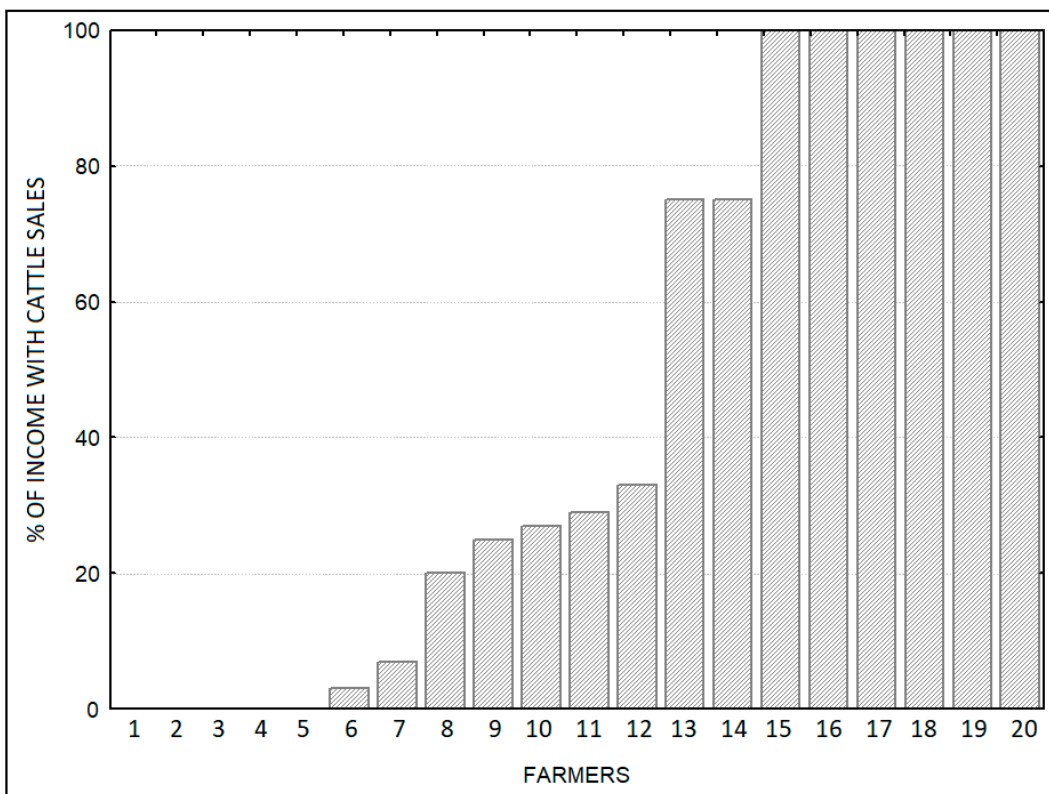

**Figure 3.** Weight of the sale of Jarmelista cattle in the farm income with cattle sales. Source: Authors' calculations from the structured survey research dataset.

**Table 3.** Characteristics of Jarmelista carcasses according to age classes.

| Age Class | | Weight (kg) | Sex | Conformation Grade | | | | | |
|---|---|---|---|---|---|---|---|---|---|
| | | | | E | U | R | O | P | TOTAL |
| 1 | <12 M | 163.64 ± 47.9 | F | | | 9 | 22 | | **31** |
| | | | M | | | 18 | 30 | | **48** |
| 2 | 13–18 M | 208.28 ± 60.1 | F | | 0 | 8 | 39 | 0 | **47** |
| | | | M | | 2 | 34 | 49 | 1 | **86** |
| 3 | 19–24 M | 247.65 ± 66.2 | F | | | 1 | 5 | 1 | **7** |
| | | | M | 1 | 4 | 13 | 24 | 0 | **42** |
| 4 | 25–36 M | 276.71 ± 76.3 | F | | 0 | 2 | 3 | 1 | **6** |
| | | | M | | 1 | 3 | 5 | 0 | **9** |
| 5 | 37–60 M | 405.46 ± 184.0 | F | 0 | 0 | 0 | 5 | 15 | **20** |
| | | | M | 5 | 5 | 10 | 0 | 0 | **20** |
| 6 | >60 M | 306.54 ± 122.0 | F | 0 | 0 | 0 | 12 | 4 | **16** |
| | | | M | 1 | 1 | 1 | 2 | 1 | **6** |
| | | | TOTAL | **7** | **13** | **99** | **196** | **23** | **338** |

Source: Breeders Association registers (Acriguarda).



Lower grades are a common feature of autochthonous breeds due to their slower growth rates, often associated with less intensive feeding. Conversely, exotic breeds have better feed conversion ratios and are mostly graded as E or U. Hence, with comparable prices for the meat, the tendency is for the reduction of the Jarmelista flock as a result of its lower profitability.

Figure 4 presents average carcass weights and the respective standard deviations of animals of different class ages. Classes 5 and 6 correspond to animals at the end of their production cycles or to animals with low reproductive performances, which were sent for slaughter. Comparing the carcass weights of groups 1–4, one can conclude about the slow growth rate of Jarmelista cattle compared to exotic breeds or even to other European autochthonous breeds within the same age classes [38]. Average weights of animals of the age class 4 (25–36 M) represent around 70% more as compared to the weights of animals in class age 1 (<12 M). This is also one reason why farmers opt to sell animals with less than 18 months (ca. 77% of age classes 1–4).

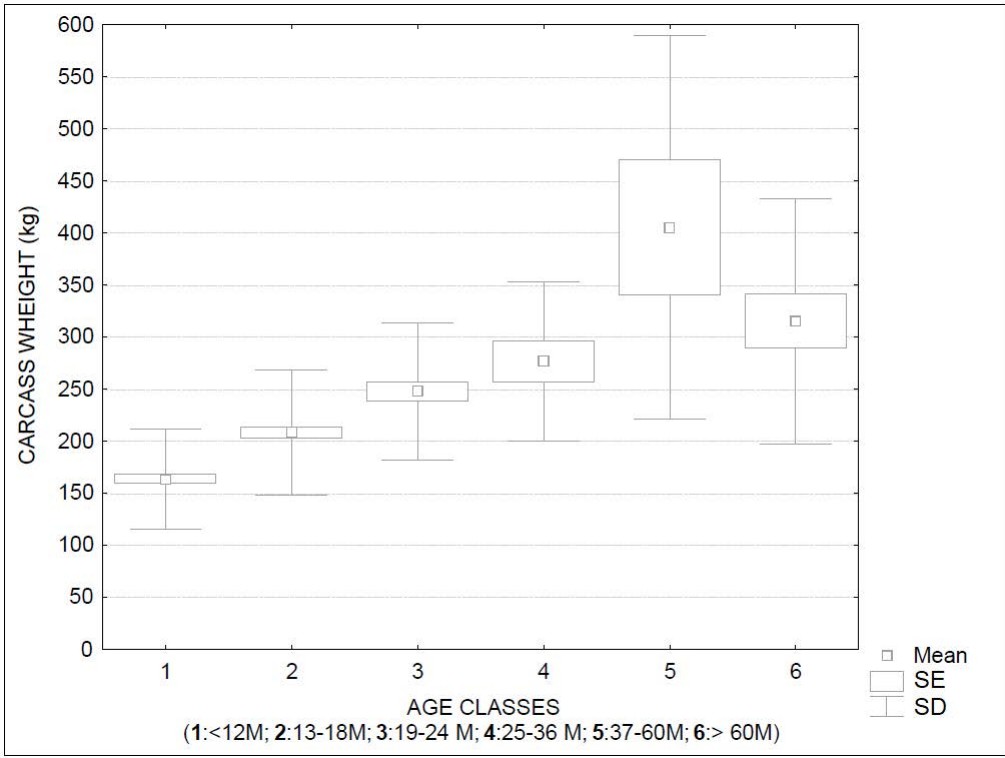

**Figure 4.** Average and standard deviation of Jarmelista carcass weights according to age classes (SE = Standard Error; SD = Standard Deviation). Source: calculation from Breeders Association registers (Acriguarda).

Polack and co-workers [39] used a multidimensional approach to evaluate the risk status of several Polish farm animal breeds. They proposed a model based on two main factors: number of females (L) and effective population size (Ne); as well as and an additional factor (D) composed of six sub-factors: (1) geographical concentration in the country; (2) demographic trend over the last 5 years; (3) cultural value; (4) parentage control; (5) ex situ conservation; (6) anthropogenic factors (breeders' organizations, financial support, activity, and age of breeders). The risk status was calculated based on the following equation:

$$X = (L + Ne + 0.5\,D)/3$$

where: *X*—risk status; L—total number of females; Ne—effective population size; D—sum of additional sub-factors.

The final score was proposed as follows: $\leq 1$—critical status; $>1$ and $\leq 2$—an endangered breed requiring action; $>2$ and $<3$—an endangered breed requiring monitoring; $\geq 3$—not at risk. Measurable factors (L and Ne) were defined according to the degree of endangerment as: critical status, endangered breed in need of conservation, endangered breed in need of monitoring, and non-endangered breed, scored as 0, 1, 2 or 3 points, respectively. Components of additional factor 3 (from D1 to D6), were scored as 1, 0.5 or 0. As the assessment of these components was subjective, this factor was weighted as 1/2 to reduce its effect on the result, and to decrease the estimation error.

By applying the abovementioned equation, the data collected by us regarding the Jarmelista breed produced a value of 1.08. Thus, although not being considered as critically endangered by using this approach, we can consider that urgent actions are required to avoid its extinction.

## 4. Discussion

Habitat destruction and degradation due to human land use for agriculture or livestock, as well as land abandonment are expected to remain the most significant cause of biodiversity loss, which allied to all environmental issues of pollution has a tremendous impact on rural areas. As global meat production and consumption is increasing, high amounts of energy and water are being used to supply the needs and, consequently, large quantities of waste and gaseous emissions are being released into the environment [40]. The sustainable use of resources and the need to balance the economic growth and the environmental preservation has become increasingly important and, as a result, the agro-industrial sector has been pressured to minimise its negative impacts in the environment.

On the other hand, autochthonous local breeds represent living, physical carriers of cultural heritage and identity in the agroecosystems of their areas of origin. Despite this, autochthonous bovine breeds, and their traditional management practices as an integral part of the rural area's life and culture, represent a low-yield farming practice that is difficult to cope with the high productivity rhythms of modern animal production. Their conservation and sustainable use are essential to conserve future breeding and livestock use and development options as well as assist to keep rural ecosystems in balance [41]. The contribution of local bovine breeds products to the local economies with benefit of the communities has been highlighted by several authors as reviewed by [42] within the BOVISOL project. These products associated with recent "local" and "slow" food trends can be recognized as high-value products as well as "good" for the planet by consumers, public administration, and scientific community.

Gandini et al. [43] reported that the degree to which a breed is exposed to becoming extinct, is an essential information to orient conservation policies. The authors proposed to estimate the number of years needed to reach a critical population size, which is also a measure of time available to evaluate options and undertake action before extinction. As reported, systematic information on the degree of endangerment for breeds can provide evidence on: (i) the erosion of breed diversity; (ii) the need to consider conservation actions for particular breeds; (iii) the urgency with which conservation strategies need to be developed and resourced; (iv) the prompting of objective breed comparisons to assess conservation value. The authors consider however that at a population size of about 100 females, probabilities of herd and population extinction increase very rapidly, and intensive population management will be required to avoid extinction. The authors report that the main problem is to define the female population size below which the breed loses its self-sustainability. This might occur due to many reasons, including the evidence that, as population size decreases, several activities intended to preserve a breed will become more difficult and/or expensive: (i) breed organization, (ii) control of performances, (iii) genetic improvement, (iv) production and commercialization of breed specific products, (v) promotion activities.

This is of utmost relevance for Jarmelista bovine breed classified as at critical risk considering the Sustainable Development Goals indicators (http://www.fao.org/dad-is/

browse-by-country-and-species/en/ (accessed on 5 January 2021)). The current efforts to undertaking risk status assessments of diversity of livestock breeds on national, regional, and global levels including the status of breeds regarding their risk of extinction, should contribute to improved effectiveness of identifying and achieving enhanced conservation measures, and should help to draw attention to socio-economic factors that can contribute to enhanced utilization of especially local or traditional breeds currently in decline, as reported by Polak and co-workers [39].

The low economic yield of this bovine breeds for farmers has to be balanced by a higher value to consumers; namely regarding the labelling of local products and creating niche markets to raise the value of local products and support sustainable, regional, small-scale production systems. In this sense, improving simultaneously the productivity of the systems and their sustainability are determinant to assure biodiversity preservation and economic and social development of the local communities, through the exploitation of existing potential for maintaining and promoting this cultural heritage and its connected knowledge systems, as well as their products, also associated to the potential use of such items to promote the regions through agro-tourism [44].

In addition, there is a growing necessity to produce quality meat, not only in its organoleptic qualities but also in regarding health warranties. The production of autochthonous breeds in Portugal can carry out these functions. Autochthonous breeds result from adaptation to often adverse environmental conditions that provide them with survival skills and disease resistance that other breeds do not have. However, in terms of productive levels, these breeds cannot compete with exotic breeds, and if they are not preserved, they may disappear. This is the case of the Jarmelista breed. Therefore, it is necessary to highlight the identity of these breeds as a determining and fundamental factor for the maintenance of the region's biodiversity through their sustainable and biological breeding and production, in addition to the valorisation of their quality attributes, by adding value to the retail price of the products in order to compensate the farmers for the low productivity of the extensive systems [20].

Garcia-Oliveira, et al. [45] recently presented a diagnosis of the challenges and problems facing the current food system to ensure a growing world population's subsistence. The first challenge is to increase production and at the same time, to introduce major transformations to obtain a sustainable food system, whose central strategy is to improve the efficiency of resource use. Agricultural systems are in constant evolution, and because of that, there is an increasing need to count on sustainable alternatives for resource use. From the challenges to sustainability, some of the most important are the rational use and management of natural resources and the promotion of sustainable production and consumption patterns. Economic growth is decoupled from pressures on ecosystems towards greater eco-efficiency of the economy and sustainable resources management. It seems that maintaining autochthonous breeds can be essential not only in the cultural context but also in a social, economic, and environmental perspective. Moreover, local food specialities can positively affect the attractiveness of tourist destinations and contribute to the regions' economic sustainability [46].

Similarly, to our study, and within a similar agro-system context, Simoncini [47] identified for the Calvana breed from Tuscany, with very high risk of extinction, the actual or potential contribution of the Alternative Food Networks to biodiversity conservation and formulated policy recommendations for local rural development. For that, this author proposed Calvana beef to be sold mainly through conventional distribution channels locally and a marketing strategy to escape producers' costs-price squeeze. It also highlighted the relevance of recognition of risk of extinction by public institutions, the need for agri-environmental measures compensating farmers to rear relics and semi-relics breeds and compulsory rearing guidelines and product labelling, linking genetic conservation to that of natural and semi-natural pastures and meadows, and potentiation of exploiting existing synergies with increasing rural tourism.

The general effectiveness of providing financial support to help autochthonous breed conservation is consensually accepted. Nevertheless, it seems essential to improve communication among breeders and other stakeholders in breed management, in order to reinforce the effectiveness of conservation measures, namely regarding the compromise between conservation and development and producer's motivation to comply with the measures adopted. In this sense, Gicquel, et al. [48] stated the need of regular analyses of conservation plans and actions to evaluate the effectiveness of the plans and improve them as necessary, based on both the resulting data and on producers' feedback.

On the other hand, land abandonment results in a negative influence on biodiversity, mainly in the mountainous areas of southern Europe [49] and underlies the role of traditional activities and traditional extensive grazing as the best practices. This will not allow to reach highest biodiversity values and the creation of a mosaic of habitats suitable for many species. Moreover, in mountainous ecosystems, there is a high risk of the loss of open habitats caused by the abandonment of long-term traditional grazing activities, which may determine the extinction of populations of species tightly linked with open habitats, namely the endemic ones.

In this way, the safeguard of Jarmelista beef market requirements and prediction is essential and justified by the increasing demand for productions of quality market trend, obtained by less intensive productions. The increased valorisation of Jarmelista beef is the main factor impacting farmers' decision to use this breed. Besides, since this breed reflects a long history of symbiosis between domestic animals and man, allied to its ecologic value, evidenced by the interaction of these animals with nature and landscape, it is urgent to take measures to protect it from extinction. Given the potential for significant future changes in production and livestock conditions, as well as regarding the protection of the natural resources, it is essential that the functions and values provided by genetic biodiversity are secured, which should be conveyed to consumers for a better and conscious valuation of sustainable products, as stated by several authors [50–53].

Therefore, the several challenges related to the valorisation of the Jarmelista breed can be summarised as follows:

(i)    the promotion of genetic preservation and sustainable production of Jarmelista breed;
(ii)   the identification of correlations between differentiating characters and biochemical/rheological profile of Jarmelista beef;
(iii)  the need to emphasise the importance of biodiversity and territorial sustainability of Jarmelista meat production;
(iv)   the development of conditions for the identification of Jarmelista beef as a differentiating element in the production and marketing of meat products;
(v)    the need to analyse the strategic positioning of Jarmelista beef in the market;
(vi)   the increase in competitiveness in the post-production value chain of Jarmelista beef;
(vii)  the development of a strategy that can support the development of Jarmelista beef products in quality records;
(viii) the development of a strategy that makes Jarmelista beef an endogenous product that enhances the economic activity and the region;
(ix)   the need to analyse the territorial differentiation factors potentiated by Jarmelista beef;
(x)    the need to increase the region's economic activity (tourism and culture);
(xi)   the promotion of cultural interest and culinary essays activities.

Therefore, a close analysis of the current situation's characterisation in the Jarmelista chain, including production, processing, trading, and marketing phases, is needed. Being a general value chain of meat production and products, as Figure 5 shows, the low economic profitability due to the lower yield of the Jarmelista breed was identified as the key obstacle in the production phase. The produced volumes were recognised as too small, even on the regional level. Partly due to low yields, the farms were not willing to increase their flocks of Jarmelista cows.

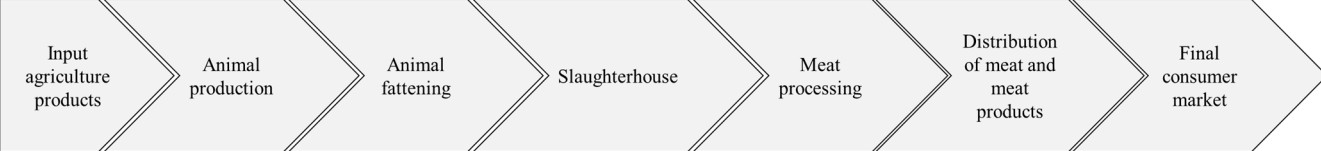

**Figure 5.** General value chain of meat production and products. Source: Adapted from Novaković, et al. [54].

The uncertainty over the availability of Jarmelista beef is seen as a problem. Thus, there are several economic constrains for the market. The undifferentiated offer of the general meat market does not help consumers choose the type of meat they acquire and consume. At the end of the value chain, in the marketing approach, much of the work must be done. Habilitating the consumers with the information of the meat characteristics and production system allows them to make conscious choices and identify and differentiate the types of meat market positioning [53,55]. This may allow consumers to understand better how to translate their ecological concerns into meat consumption. In this sense, both claims, mountain origin and autochthonous breed, convey relevant information to consumers, as well as the environmental setting. However, these claims are not perceived by the market.

The main ideas of these perspectives, identified within this study, can be summarised in Table 4, and perceived as positive and negative, i.e., opportunities or challenges to be addressed.

**Table 4.** Situation analysis for Jarmelista breed chain different perspectives.

|  | Ecological | Social | Cultural | Economic |
|---|---|---|---|---|
| **Positive aspects** | Characteristics and versatility of animals | Fixing population in rural areas | Preservation of an autochthonous breed | Improving business competitiveness of Jarmelista meat production |
|  | Production system | Improving population identity | Preservation of traditional production methods | Improving market satisfaction |
|  | Biodiversity preservation of the territory | Preservation of territory's landscape | Meat is an appreciated cultural product of the Portuguese diet | Enhancing new business creation in tourism and restauration sector |
|  | Better response to consumer ecological concerns |  | Enhancing traditional cuisine |  |
|  | Consumers interest in alternative, less intensive, forms of production. |  |  |  |
| **Negative aspects** | Ecological arguments for not sufficient eco achievements of meat production | Low population density area | Difficulties in cultural and traditional proud arguments | Business knowledge capacity |
|  | Low ecological education and knowledge | Typical difficulty for farmers association in the country |  | Marketing perspective adoption |
|  |  |  |  | Low economic capacity |
|  |  |  |  | Consumer willingness to pay for quality products |

Source: Authors' analysis from the research dataset.

This first identification must be deepened, as it seems crucial for the Jarmelista beef value chain's overall improvement. It already allows highlighting several important constraints to achieve the goals of the "Valor Jarmelista Project" (i.e., territory preservation and valorisation, the safeguard of the autochthonous breed, and the valorisation of Jarmelista beef production and marketing). More profound research in the consumer perspective must be engaged, and an identification of the meat characteristics must be specified, but overall, we may say that the Jarmelista meat production system has potential to be increased and improved. It is expected that, as a result of the actions undertaken during the execution of the Valor Jarmelista Project, the main positive impact to be assessed afterwards should be a significative increase in the numbers of Jarmelista females used for breeding purposes. This impact will be evaluated by the end of 2021. However, taking into consideration the data already available, expectations are low. If the different stakeholders associated with the production chain fail to reverse the actual status, most probably, the breed will be lost in the short future. We recommend therefore that public authorities should undertake additional specific actions in order to preserve the breed. These actions should encompass a specific ex-situ conservation programme and parentage control, associated to better economic incentives to breeders than the ones currently available.

## 5. Conclusions and Implications

The increasing awareness of the population about the use of natural resources in a sustainable way, ensuring its use by the present and future generations, has been the main concern for successive agricultural policies to preserve plant and animal genetic resources. Furthermore, as described previously, agriculture is essential for sustainable development in the Guarda region in their economic, social, and environmental aspects. Jarmelista animals are characterised by being rustic and perfectly adapted to all the unfavourable conditions of the region. The Jarmelista breed, being a recognised Portuguese autochthonous breed with specific qualities and attributes, such as biological breeding and sustainable production system, needs to be adequately valued and integrated in the growing meat market for products of quality market trend, obtained by less intensive production systems. Besides, it is mandatory to provide adequate support to protect Jarmelista breed from extinction considering the ecologic value and providing an oriented market strategy based on a better and conscious valuation of sustainable Jarmelista meat.

In this sense, it is crucial to shift this breed as a marginal complement to a relevant and profitable income source for local producers. This goal can only be achieved if the different stakeholders involved in the value chain can work together and transform the main challenges identified in opportunities for the region.

The consumer's trends on sustainable products, which are also linked to healthier products, are consistent with this type of beef production. To increase their market value, these features must be clearly communicated. The small producer does not have the adequate skills and financial capacity to do it. Therefore, marketing campaigns led by the local government institutions is crucial.

If the strategy based on the increase of the profitability of Jarmelista meat fails to invert the declining numbers of the breed, authorities must act putting in practice active measures to preserve it. Of particular interest could be the promotion of "official farms" dedicated to in situ conservation of this and of other menaced breeds. These farms should be controlled by government authorities and be focused on the maintenance of adequate numbers of specimens as well as on the demonstration of best production practices. This strategy has already been used in the past but has been abandoned by reasons grounded on lean management applied to the public administration what, in our opinion, was a major error. This research must be continued in the business and market analysis perspective to fully understand the possibilities that are here designed to prevent the extinction of Jarmelo breed.

**Author Contributions:** Conceptualization, P.C., T.P.; methodology, P.C. and T.P.; validation, P.C., T.P. and C.P.; formal analysis, P.C., T.P., and. C.P.; investigation, P.C., C.P and T.P., and.; resources, P.C., C.P and T.P.; writing—original draft preparation, T.P., P.C., M.S. and C.P.; writing—review and editing, P.C., T.P. and C.P.; project administration, T.P. and C.P.; funding acquisition, T.P. and C.P. All authors have read and agreed to the published version of the manuscript.

**Funding:** This work was supported by national funds through Agriculture and Rural Development's ministry and co-financed by the European Agricultural Fund for Rural Development (EAFRD), through the partnership agreement Portugal2020—PDR, under the project PDR2020-101-030748: Valor Jarmelista.

**Institutional Review Board Statement:** Not applicable for studies not involving humans or animals.

**Informed Consent Statement:** Informed consent was obtained from all subjects involved in the study.

**Data Availability Statement:** The data presented in this study are available on request from the corresponding author. The data are not publicly available due to respect of privacy rules.

**Conflicts of Interest:** The authors declare no conflict of interest and the funders had no role in the design of the study; in the collection, analyses, or interpretation of data; in the writing of the manuscript, or in the decision to publish the results.

## Appendix A. Survey of Cattle Farms of the Jarmelista Breed

1.  Farm name: ___________________________________________________
2.  Location of the farm: _____________
3.  What weight does the production of Jarmelista cattle have on cattle sales?

    ○   <25%
    ○   [25:50]%
    ○   [51:75]%
    ○   >75%

4.  Indicate the allocation of the area of your holding to the different activities:

|  | Area (ha) |
|---|---|
| Total surface area | |
| Used Agricultural Area (UAS) | |
| Pastures | |
| Annual crops for livestock | |

5.  Indicate the actual number of animals on your farm:

|  | Number of Animals Jarmelista Breed/Other Breeds |
|---|---|
| Calves up to 6 months | |
| Male calves from 6 to 12 months | |
| Females calves from 6 to 12 months | |
| Steers from 12 to 24 months | |
| Heifers from 12 to 24 months | |
| Bulls | |
| Cows | |

6.  Indicate, for the last year of production:

| Animals | | Number of Animals Jarmelista Breed/Other Breeds | | |
|---|---|---|---|---|
| | | **Number of Animals** | **Date** | **Post-Weaning Feed to Slaughter** |
| Young | Born | | | |
| | Weaned | | | |
| Adults | Replacement | | | |
| | Scrap | | | |
| Dead | At Birth | | | |
| | Young | | | |
| | Adults | | | |

7.  Indicate the results of reproductive management for Jarmelista breed

| |
|---|
| Average age at 1st Delivery |
| Average interval between Deliveries (only for the last delivery) |
| Average Interval between Total Deliveries |

8.  Feed characterization. Please indicate the main types of feeds used in the farm (e.g., grazing; straw; commercial feed; fodder; other)
9.  Farm Characterization

| | **Alive** | **For Slaughter** |
|---|---|---|
| Number of total animals sold | | |
| Total number of Jarmelista animals sold | | |
| Total volume of animal sales | | |
| Sales volume of Jarmelista animals | | |

Calculation of risk status of a breed (Based on Polack and co-workers [39]).

| **X = (L + Ne + 0.5D)/3** | **Points** |
|---|---|
| **X = 1.08 = (2 + 0 + 0.5*(1+0.5+0.5+0+0+0.5))/3** | |
| X—risk status; | |
| L—total number of females; | |
| Ne—effective population size; | |
| D—sum of additional elements (sub-factors). | |
| The final score, results in risk status assessment, is as follows: ≤1—critical; > 1 and ≤2—an endangered breed requiring action; > 2 and <3—an endangered breed requiring monitoring; ≥ 3—not at risk. | |
| Components of additional factor 3 (from D1 to D6), were scored as 1, 0.5 or 0. As the assessment of these components was subjective, this factor was weighted as 1/2 to reduce its effect on the end result, and to decrease the estimation error. | |
| L = 1 < 150 females | |
| L = 2 150–1000 females | **2** |
| L = 3 > 7500 females | |
| L = 4 > 25000 females | |
| Effective population size: Ne = 4 NMNF/(NM + NF) | |
| *NM = the number of males* | |
| *NF = the number of females* | |

| | |
|---|---|
| Effective population size is a key parameter for describing genetic diversity in animal populations and predicting rate of inbreeding. Correction to random selection of males and females for mating was applied based on Santiago and Caballero (1995), where: $N_e$ = original $N_e \times 0.7$. The following thresholds were adopted: $N_e \leq 50$—0 points, critical status; $50 < N_e \leq 200$;—1-point, endangered breed in need of conservation; $200 < N_e \leq 1000$;—2 points, endangered breed in need of monitoring, and $N_e > 1000$—3 points, non-endangered. It was also assumed that regardless of the final score, if $N_e \leq 50$, a breed is considered endangered and in need of conservation, as there is a direct and inversely proportional relationship between the effective population size and the rate of inbreeding. **$N_e = (4*10*161)/(10 + 161)*0{,}7 = 26.36$** | **0** |
| D1—geographical concentration in a country. We adopted after Alderson (2010), a score scale where 1 point was awarded when $\geq 75\%$ of the population was concentrated in the region of origin, 0 if only $\leq 25\%$ of the breed occurred in the region of origin, 0.5 points if the breed concentration was intermediate. Where the population occurred in a small number of herds ($\leq 2$), regardless of the end results, the breed was considered endangered and in need of conservation. | **1** |
| D2—the demographic trend over the last 5 years. This is an important and objective factor indicative of the population's development. Three scores were awarded: 1 point—upward trend; 0.5—stable trend; and 0—downward trend. | **0.5** |
| D3—the cultural value of a breed (documented links with tradition, culture, and the region): 1 point—has cultural/historical value; 0.5—little cultural/historical value; 0—no cultural/historical values. Points awarded by experts of the species-specific Working Groups—that serve as advisory bodies to the National Research Institute of Animal Production. Some breeds have been present in Poland for centuries and are an important part of the country's cultural heritage. The value of the breed was assessed on the basis of specific elements, such as it being used locally directly or indirectly in contributing to handicraft, folklore, artistic expression, and religious traditions, it being used in local gastronomy and products, and any roles provided in maintaining a specific landscape. The scoring was as follows: 1 point—no cultural/historical value; 0.5—little cultural/historical value; 0—has cultural/historical value. | **0.5** |
| D4—parentage control: 1 point—present; 0.5 –present to a small extent; 0—absent | **0** |
| D5—*ex situ* conservation: 1 point—present; 0.5—present to a small extent; 0—absent | **0** |
| D6—anthropogenic factors, assessed by experts and on the basis of age of breeders, their activity in implementation of existing conservation programmes (e.g., participation in exhibitions, popularization of breeds and their products, collaboration with breeding organizations), and possibility of financial support for breed conservation: 1 point—anthropogenic factors present; 0.5—partly present; 0—absent. | **0.5** |

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
