# Peer review of "Sustainable Local Exploitation and Innovation on Meat Products Based on the Autochthonous Bovine Breed Jarmelista"

_sustainability, doi:10.3390/su13052515_

Round 1

Reviewer 1 Report

Dear authors,

Your article is about sustainable local exploitation and innovation on meat products. Based on the production of local meat (Jarmelista breed cattle), your article is captivating and very descriptive.

However, there are some question marks about your research:
- what is the novelty of your article?
- it is necessary to explain some important concepts, such as: sustainable local exploitation, innovation on meat products, sustainable production, and others, in a chapter about "Theoretical Background" or "Literature review".
- What are the research questions in your analysis? or What are the research hypotheses in your analysis?
- in the discussion chapter it is important to add the results obtained, either to the research questions or to the testing of the research hypotheses.
- there is a need for additional explanations in the research methodology and to refer to an annex with data collected from the 20 farms included in the analysis.
- what is the source of the data from table 1? This table can be included in the annexes.
- what is the source of the data from table 2?
- how was the equation on page 10 calculated? What are the results obtained?
- what is the source of the data from row 477-494?
- what is the source of the data from table 4? How was this table designed? It is not very clear.
- what are the limits of research?
- what is the importance of the study?

These are just some of the issues that need clarification.

Good luck! 

Author Response

We wish to thank the reviewers for going through our manuscript and making valuable comments to further improve the article's quality. We have addressed all reviewer's comments and hope that the revised version is found suitable for publication in the Journal Sustainability - Special Issue "Sustainable Rural Development through Entrepreneurship and Innovation”.

Reviewer 2 Report

The statistical data referred to must be updated. The year 2004 is quite far from 2021 and I don't think it is eloquent. The years for which the data presented in point 3.1 are valid are not mentioned. Better argumentation for supporting the breeding of these breeds of bovines (Jarmelista) compared to other breeds (comparative). Insisting on concrete policies and possible strategies to save this breed of bovines.  The arguments for integrating race into sustainable development need to be deepened. There is too much talk about race and less about the role of this race in sustainable development.  It is also useful cartography with the study area and the positioning of the farms. 

Author Response

We wish to thank the reviewers for going through our manuscript and making valuable comments to further improve the article's quality. We have addressed all reviewer's comments and hope that the revised version is found suitable for publication in the Journal Sustainability - Special Issue "Sustainable Rural Development through Entrepreneurship and Innovation”.

The remarks of reviewers are retyped below in normal font, and our responses and the actions that we took to address the comments are in blue normal font.

The statistical data referred to must be updated. The year 2004 is quite far from 2021 and I don't think it is eloquent.

We appreciate the reviewer's feedback. To the best of our knowledge, this work is the most recent report on sustainable production in Portugal and broadly cited in different academic and scientific documents regarding this subject. Nevertheless, it was included more recent data from Willer, H., Schlatter, B., Trávníček, J., Kemper, L., & Lernoud, J. (2020). The world of organic agriculture. Statistics and emerging trends 2020.

The years for which the data presented in point 3.1 are valid are not mentioned.

Thank you for pointing this out. The years of the data presented are refereed in the references cited and in the text.

Better argumentation for supporting the breeding of these breeds of bovines (Jarmelista) compared to other breeds (comparative).

Thank you for this suggestion. Some information was included in the text (lines 280-289)

Insisting on concrete policies and possible strategies to save this breed of bovines.

Thank you for the comment. Conclusions section was renamed as Conclusions and Implications and modified highlighting the implications and utility of this study for policy makers.

The arguments for integrating race into sustainable development need to be deepened. There is too much talk about race and less about the role of this race in sustainable development. 

Thank you for the comment.

The production of this specific breed is described, and we reinforced the sustainable perspective – Point 3.3 Characterization of the Production System (lines 272-289)

It is also useful cartography with the study area and the positioning of the farms.

The Map of the region Jarmelita farms was included

Reviewer 3 Report

Thank you for the opportunity of reviewing your manuscript. It addresses a topic of high importance and within the journal s scope. The paper presents and discusses the results of an empirical research. The methodology is adequate and the paper is well written. My only concern regards the implications/theoretical and practical utility/policy of the paper, and I suggest that the authors address this issue in a separate section or within the final Conclusions section, which should be then renamed as Conclusions and implications. One other shortcoming regards the relatively short list of references, and authors should review the list and add / reference more if case. Good luck!

Author Response

We wish to thank the reviewers for going through our manuscript and making valuable comments to further improve the article's quality. We have addressed all reviewer's comments and hope that the revised version is found suitable for publication in the Journal Sustainability - Special Issue "Sustainable Rural Development through Entrepreneurship and Innovation”.

The remarks of reviewers are retyped below in normal font, and our responses and the actions that we took to address the comments are in blue normal font.

Thank you for the opportunity of reviewing your manuscript. It addresses a topic of high importance and within the journal s scope. The paper presents and discusses the results of an empirical research. The methodology is adequate, and the paper is well written.

We thank the reviewer's feedback and positive comments.

My only concern regards the implications/theoretical and practical utility/policy of the paper, and I suggest that the authors address this issue in a separate section or within the final Conclusions section, which should be then renamed as Conclusions and implications.

Thank you for pointing this out. Conclusions section was renamed as Conclusions and Implications and modified highlighting the implications and utility of this study for policy, according to reviewer’s suggestions.

One other shortcoming regards the relatively short list of references, and authors should review the list and add / reference more if case. Good luck!

Thank you for this suggestion. Considering the reviewer's comments and suggestions, the reference list was modified because of the new data and references to other studies and reports.

Round 2

Reviewer 1 Report

Dear authors,

Thank you very much for the comprehensive answers and for improving your article.

Good luck!